



# Empirical Bayes approach to climate model calibration

Charles S Jackson[1] and Gabriel Huerta[2]

[1]Institute for Geophysics, University of Texas at Austin, Austin, TX, USA
[2]Department of Math and Statistics, University of New Mexico, NM, USA

*Correspondence to:* Charles Jackson (charles@ig.utexas.edu)

**Abstract.** Climate data is highly correlated through the physics and dynamics of the atmosphere. Model evaluation often involves averages of various quantities over different regions and seasons making it difficult from a statistical perspective to quantify the significance of differences that arise between a model and observations. Here we present a strategy that makes use of a set of perfect

modeling experiments to quantify the effects of these correlations on model evaluation metrics. This information is incorporated into Bayesian inference through a precision parameter with informative priors. These concepts are illustrated through an example of fitting a line through data that includes either uncorrelated or correlated noise as well as to the calibration of CAM3.1. The concept of a precision parameter may be applied as a strategy to weight different climate model evaluation metrics

within a multivariate normal framework. From the example with CAM3.1, the precision parameter plays a central role in rescaling the estimated parametric uncertainties to better accommodate modeling structural errors.

## 1 Introduction

Within Bayesian inference, calibration refers to the problem of estimating model parameters given

observations and uncertainties that affect model-data differences. The result of a calibration is a joint probability distribution whose maximum density identifies the region of parameter space where model performance is optimal in the sense that a weighted sum of distance between the model and observations is minimized. Among the challenges that exist for Bayesian calibration of climate models is the question of how to assign weights to various observational targets, many of which are

highly correlated through the physics and fluid dynamics of the atmosphere. Here we present and discuss the use of a precision parameter which is a common device in Bayesian calibration to make use of information concerning how data is scattered about the model to better quantify uncertainties affecting model calibration.

  One of the issues that needs to be considered in the application of a precision parameter to climate

model calibration is that uncertainties affecting model evaluation are high dimensional. A scalar will have limited scope to compensate for information that is missing within an evaluation metric



concerning the many space and field dependencies that exist in the data. Thus one of our objectives here is to show both the usefulness and limitations of using a scalar parameter to assess climate model parametric uncertainties.

Ideally, dependency information is included within model evaluation metrics as it would provide an excellent way to determine whether a climate model is capturing observed climate phenomena for the right reasons. However, it is still difficult to represent such dependency information mathematically (Mu et al., 2004). Here we explain how the use of a precision parameter can be enhanced by additional information that can be obtained from a set of perfect modeling experiments in which the

only errors producing differences between a model and synthetic 'observations' come from internal variability. This inherently empirical approach allows us to assess the significance of the distance between a model and data even though we don't have a perfect representation of dependency information. This circumstance is perhaps unique to climate models which are able to represent important aspects of the chaotic motions of the fluid atmosphere and ocean and processes affecting the radia-

tive transport of energy between the top of the atmosphere and surface. Indeed, it would be difficult to design a more sophisticated statistical model of climate modeling errors than climate models themselves.

Sections 2 and 3 provides background material concerning climate model calibration. While not new, this material may be unfamiliar to physical scientists and will be important to the understanding

and application of ideas concerning use of idealized modeling experiments to help establish informative priors to help deal with the sometimes arbitrary ways models are tested against observational data.

## 2    Climate model evaluation metrics

### 2.1    Multivariate normal metric

In order to explain the effects of dependences on parameter uncertainties, consider a multivariate normal statistical model for assessing the significance of the distance between the output of a simulation with two observables $d_1$ and $d_2$. Suppose $\hat{x}_1$ is an estimate of observations $d_1$ using climate model $g(\mathbf{m})$ with parameters $\mathbf{m}$. If errors are normally distributed, which is a good assumption for monthly mean or longer climate data, $\hat{x}_1 = d_1 + \epsilon_1$, where $\epsilon_1$ is the model errors with $\epsilon_1 \sim N(0, \sigma_1^2)$. $\epsilon_1$ in-

cludes both the modeling error due to the grid resolution, the possibility of unparameterized physical processes (i.e. missing physics), uncertainty in the initial and surface forcing conditions, and the instrumental error. Many of these types of errors are hard to estimate particularly if the process that generated them are correlated in space and time.





If it is assumed that $\epsilon_1$ is distributed as a Gaussian with variance $\sigma_1^2$, the likelihood of simulating
data $d_1$ with the model $g(\mathbf{m})$ using parameters $\mathbf{m}$ is given by

$$\theta(d_1|\hat{x}_1, \mathbf{m}) = \frac{1}{\sigma_1\sqrt{2\pi}} exp\Big(-\frac{(\hat{x}_1 - d_1)^2}{2\sigma_1^2}\Big). \tag{1}$$

In the case where the model simulates *correlated* observations $d_1$ and $d_2$, the joint likelihood for
simulating this data given the same model can be expressed as

$$\theta(d_1, d_2|\hat{x}_1, \hat{x}_2, \mathbf{m}) = \frac{1}{\sigma_1\sigma_2 2\pi(1-\rho^2)} exp\Big(-\frac{1}{2(1-\rho^2)}\Big[\frac{(\hat{x}_1 - d_1)^2}{\sigma_1^2} +$$
$$\frac{(\hat{x}_2 - d_2)^2}{\sigma_2^2} - 2\rho\frac{(\hat{x}_1 - d_1)(\hat{x}_2 - d_2)}{\sigma_1\sigma_2}\Big]\Big), \quad (2)$$

where $\rho$ is the correlation coefficient of observables $d_1$ and $d_2$. Note that if $\rho = 0$, then $\theta(d_1, d_2|\hat{x}_1, \hat{x}_2, \mathbf{m})$
becomes the product of the two Gaussian distributions $\theta(d_1|\hat{x}_1, \mathbf{m})$ and $\theta(d_2|\hat{x}_2, \mathbf{m})$ which is what
we would get if $d_1$ and $d_2$ were independent. Note that the probability density is not defined for
the case where the observations are perfectly correlated, $\rho = 1$. Written in matrix notation, the joint
probability distribution for *correlated* observables $d_1$ and $d_2$ is

$$\theta(\mathbf{d}|\mathbf{x}, \mathbf{m}) = \frac{1}{2\pi|\mathbf{C}|^{\frac{1}{2}}} exp\Big(-\frac{1}{2}(\mathbf{x} - \mathbf{d})^T\mathbf{C}^{-1}(\mathbf{x} - \mathbf{d})\Big) \tag{3}$$

where

$$\mathbf{x} = \begin{bmatrix} \hat{x}_1 \\ \hat{x}_2 \end{bmatrix}, \quad \mathbf{d} = \begin{bmatrix} d_1 \\ d_2 \end{bmatrix}, \text{ and } \quad \mathbf{C} = \begin{bmatrix} \sigma_1^2 & \rho\sigma_1\sigma_2 \\ \rho\sigma_1\sigma_2 & \sigma_2^2 \end{bmatrix}. \tag{4}$$

The definition for $k$ observations is

$$\theta(\mathbf{d}|\mathbf{x}, \mathbf{m}) = \frac{1}{\sqrt{(2\pi)^k|\mathbf{C}|}} exp\Big(-\frac{1}{2}(\mathbf{x} - \mathbf{d})^T\mathbf{C}^{-1}(\mathbf{x} - \mathbf{d})\Big), \tag{5}$$

which only works mathematically if $\mathbf{C}^{-1}$ exists. Since many observations of climate are strongly
correlated there is a good chance that the covariance matrix is rank deficient, its inverse is singular,
and its determinant is 0. The typical solution to this problem is to apply singular value decomposition
to $\mathbf{C}$ to identify a reduced number $k_e$ of orthogonal dimensions commonly referred to as empirical
orthogonal functions or 'eofs' in the atmospheric sciences (e.g., Mu et al., 2004). The argument of
the exponent, when rotated into this orthogonal basis and truncated to include the first $k_e$ vectors
associated with the largest eigenvalues, follows a $\chi^2_{(k_e)}$ distribution with $k_e$ degrees of freedom

$$\sum_{i=1}^{k_e} \frac{[\mathbf{eof}_i^T(\mathbf{x} - \mathbf{d})]^2}{\sigma_i^2} \sim \chi^2_{(k_e)} \tag{6}$$

If the model is unbiased, the $\chi^2_{(k_e)}$ distribution in this case will have an expected mean of $k_e$ and a
variance of $2k_e$. Thus within a multivariate normal framework for testing a model against a set of
observational targets, the average value of the argument of the exponent within equation (3), which





scales with the effective degrees of freedom, is an important determinant in the width of estimated confidence intervals. Often climate model performance metrics involve averages over different regions, seasons, and quantities, making estimates of parametric uncertainty arbitrary without some way to incorporate a scale in which to judge changes in model skill relative to a null hypothesis.

## 3 Model calibration with precision parameter $S$

We now introduce a precision parameter $S$ that can scale an arbitrarily defined climate model evaluation matrix (Jackson et al., 2008). Statisticians often include a similar parameter within Bayesian calibration as one can use information about how well a model matches data to determine uncertainties in parameters. One can think of it as representing uncertainties in the specification of $\mathbf{C}^{-1}$. Simply scaling $\mathbf{C}^{-1}$ can not affect the relative weighting between different observational targets. However such a factor can affect the presumed strength of the observational evidence, similar to changing the effective degrees of freedom, such that solution probabilities are narrower or larger depending on the size of model-data mismatch. By introducing $S$ as an uncertain parameter in addition to climate model parameters $\mathbf{m}$ we need to consider a distribution for representing its 'prior' uncertainty. Moreover, because of the relationships between $S$ and the covariance matrix, the value of $S$ needs to scale with the errors that exist between a model and data. The point of this section is to describe the selection of a prior distribution for $S$ and to provide guidance on how its co-dependency with model parameters $\mathbf{m}$ can be estimated through a hybrid of Metropolis and Gibbs sampling (Gelman et al., 2013).

The Bayes expression for estimating a set of parameters $\mathbf{m}$ and parameter $S$ conditional on observations $\mathbf{d}_{obs}$ is

$$p(\mathbf{m}, S|\mathbf{d}_{obs}) \propto l(\mathbf{d}_{obs}|\mathbf{m}, S)p(\mathbf{m}, S) \tag{7}$$

We start by assuming the priors for $\mathbf{m}$ and $S$ are independent, i.e. $p(\mathbf{m}, S) = p(\mathbf{m})p(S)$. One choice for prior $p(S)$ is the gamma distribution, $p(S) \sim Ga(\alpha, \beta)$ with shape parameter $\alpha > 0$ and rate parameter $\beta > 0$,

$$p(S) = \frac{\beta^{\alpha}}{\Gamma(\alpha)} S^{\alpha-1} exp(-\beta S). \tag{8}$$

This choice of the functional form for $S$ is convenient as it is conditionally conjugate to the multivariate normal formulation of the likelihood function, which facilitates the use of Gibbs sampling for iteratively estimating co-dependencies between model parameters $\mathbf{m}$ and $S$ within the likelihood. The $Ga(\alpha, \beta)$ distribution appears as a skewed distribution with a mean and variance provided by

$$<S> = \frac{\alpha}{\beta} \tag{9}$$

and

$$var(S) = \frac{\alpha}{\beta^2} \tag{10}$$





With the proposed gamma distribution for $S$, the distribution for the likelihood function, assuming errors are multivariate normal, is

$$l(\mathbf{d}_{obs}|\mathbf{m},S) \propto S^{\frac{k_e}{2}} exp(-SE(\mathbf{m})) \tag{11}$$

where $E(\mathbf{m})$ is the metric of climate model performance, which for a multivariate normal errors is the argument of the exponential given in equation (5)

$$E(\mathbf{m}) = \frac{1}{2}(g(\mathbf{m}) - \mathbf{d}_{obs})^T \mathbf{C}^{-1}(g(\mathbf{m}) - \mathbf{d}_{obs}). \tag{12}$$

The term $S^{\frac{k_e}{2}}$ is the pre-exponential factor for a Gaussian distribution which due to uncertainties in $S$ can no longer can be assumed to be constant. One may think of $S^{\frac{k_e}{2}}$ as a scalar factor affecting $|\mathbf{C}|^{\frac{1}{2}}$ in equation (5) which includes $S$ as the factor affecting the precision of each of the $k_e$ independent observations

$$\mathbf{C} = \begin{bmatrix} \frac{1}{S} & & 0 \\ & \ddots & \\ 0 & & \frac{1}{S} \end{bmatrix}_{[k_e \times k_e]}, \tag{13}$$

such that

$$|\mathbf{C}|^{\frac{1}{2}} = (\frac{1}{S})^{\frac{k_e}{2}}. \tag{14}$$

An important implication for assuming independence between priors for $\mathbf{m}$ and $S$ is that we now can use Gibbs sampling to iteratively estimate their co-dependency using two separate equations:

$$p(\mathbf{m}|S,\mathbf{d}_{obs}) \propto l(\mathbf{d}_{obs}|\mathbf{m},S)p(m)$$
$$\propto exp(-SE(\mathbf{m}))p(m) \tag{15}$$

and

$$p(S|\mathbf{m},\mathbf{d}_{obs}) \propto l(\mathbf{d}_{obs}|\mathbf{m},S)p(S)$$
$$\propto S^{\frac{k_e}{2}+\alpha-1} exp(-S[E(\mathbf{m})+\beta]) \tag{16}$$

The last expression for equation (16) comes from the the product of equation (11) and equation (8), omitting constant factors. One can iteratively generate a value of $\mathbf{m}$ conditional on $S$ and a value of $S$ conditional on $\mathbf{m}$ in the following way:

1. To simulate $\mathbf{m}$ conditional on $S$, apply a Markov Chain sampling algorithm for $\mathbf{m}$ (equation 15) but just one iteration.

2. To simulate $S$ conditional on $\mathbf{m}$, sample from a gamma distribution (equation 16) which has scale parameter $\frac{k_e}{2}+\alpha$ and rate parameter $E(\mathbf{m})+\beta$. The values for $k_e$, $\alpha$, and $\beta$ are specified according to the empirical Bayes principles outlined in the following section.





3. Repeat steps 1 and 2 several times until convergence is achieved.

According to this sampling strategy, the mean value for $S$ conditional on $\mathbf{m}$ will be

$$< S >= \frac{\frac{k_e}{2} + \alpha}{E(\mathbf{m}) + \beta}. \tag{17}$$

The mean value for $S$ decreases with increasing errors between the model and data. As $S$ decreases, the likelihood function becomes less discriminating of alternate models and estimates of confidence intervals will increase in range. As will become clear in the example, the increase in range is not arbitrary. It is precisely what is needed to account for any errors in the description of uncertainty in the climate model metric as given by $E(\mathbf{m})$, at least when the model is 'perfect'.

## 4 Empirical Bayes


The sampling of $S$ within equation (16) depends on parameters $k_e$, $\alpha$, and $\beta$. While it is always necessary to provide information about $k_e$, it is common to select non-informative values for $\alpha$ and $\beta$, such as $\alpha = 0$ and $\beta = 0$, which would allow posterior estimates of $S$ to be dominated by information about model-data misfit coming from the log-likelihood (equation 12). As climate model
metrics often include multiple observational targets and quantities that are averaged over different regions and seasons, we propose a process to make use of a set of 'perfect' modeling experiments to provide additional information about how all these different quantities could be weighted using $S$. A 'perfect' model is one where we replace model output for observational data. Repeated experiments with different initial conditions explores the effects of internal variability on the metric of model
errors that is used within the likelihood function. Since we are using data to inform the prior, this is generally referred to as empirical Bayes. Sections 5 and 6 will apply empirical Bayes to the example of fitting a line to data and discuss the application to climate model calibration.

### 4.1 Effective degrees of freedom $k_e$

Here we determine the effective number of degrees of freedom for a set of experiments in which we
use a climate model to simulate the effects of correlated noise on an evaluation metric. We start by assuming that the evaluation metric will be proportional to a $\chi^2_{(k_e)}$ distribution with $k_e$ degrees of freedom,

$$E(\mathbf{m}) = \frac{A}{2} \chi^2_{(k_e)}, \tag{18}$$

where $A$ is an unknown constant. $E$ is related to degrees of freedom $k_e$ by $< E >= \frac{Ak_e}{2}$ and
$var(E) = \frac{A^2 k_e}{2}$ where the angle brackets denote an ensemble mean and $var(E)$ is the ensemble variance using $N_{exp}$ number of perfect modeling experiment samples. It follows by substitution that

$$k_e = \frac{2 < E >^2}{var(E)}. \tag{19}$$



This estimate is not very precise, especially without a lot of samples. For example, Figure (1) indicates there is about a 30% error in estimates of $k_e$ using $N_{exp} = 100$ samples. If $k_e$ for these 100 samples were estimated to be 21, then the actual $k_e$ of the data could as low as 14 or as high as 28. Below we incorporate the uncertainty in the estimates of the degrees of freedom into the priors for the precision parameter $S$.

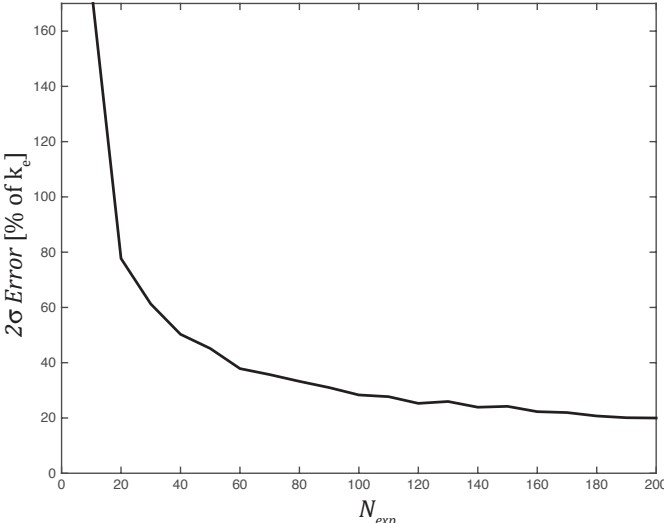

**Figure 1.** Uncertainty in the estimate of the effective degrees of freedom using equation(19) as a function of the number of perfect modeling experiment samples $N_{exp}$. The $2\sigma$ uncertainty is expressed as a percentage of $k_e$.

### 4.2 Scale and rate parameters $\alpha$ and $\beta$

Using $N_{exp}$ perfect modeling experiments, one may estimate values for the gamma distribution parameters $\alpha$ and $\beta$. The goal is to select values for $S$ that would allow $N_{exp}$ samples of $SE(\mathbf{m})$ to generate the same mean and 95% credible interval as a $\frac{1}{2}\chi^2_{(k_e)}$ distribution with $k_e$ degrees of freedom,

$$< S >= \frac{\alpha}{\beta} = \frac{\sqrt{\frac{1}{2}k_e}}{\sigma_{E(\mathbf{m})}} \tag{20}$$

where $\frac{1}{2}k_e$ is the variance of $\frac{1}{2}\chi^2_{(k_e)}$ and $\sigma_{E(\mathbf{m})}$ is the standard deviation of $E(\mathbf{m})$ using $N_{exp}$ samples. The variance in $S$ is based on the uncertainty in estimating $\sigma_{E(\mathbf{m})}$ with only $N_{exp}$ samples,

$$var(S) = \frac{\alpha}{\beta^2} = var(\tilde{S}^{N_{exp}}) < S >^2, \tag{21}$$





where samples of $\tilde{S}^{N_{exp}}$ are generated according to

$$\{\tilde{S}^{N_{exp}}\}_i = \left\{ \frac{\sqrt{\frac{1}{2}k_e}}{\{\sigma_{\chi^2_{(k_e)}}\}^{N_{exp}}} \right\}_i. \tag{22}$$

Here $\{\sigma_{\chi^2_{(k_e)}}\}^{N_{exp}}$ is an estimate of $\sigma_{\chi^2_{(k_e)}}$ using $N_{exp}$ samples. $\{\sigma_{\chi^2_{(k_e)}}\}^{N_{exp}}$ may be estimated any number of times. Note that $<\tilde{S}^{N_{exp}}> \sim 1$ because $\tilde{S}$ represents the scaling parameter for a cost function in which the effective degrees of freedom is known. The expression for $\alpha$ ends up being independent of the perfect modeling experiments,

$$\alpha = \left[ var(\tilde{S}^{N_{exp}}) \right]^{-1}. \tag{23}$$

## 5 Example: fitting a line

### 5.1 Uncorrelated noise

Suppose one has a linear model of some data $\mathbf{d}_{obs}$ taken at 100 points $\mathbf{x}$.

$$d_i = ax_i + b + \epsilon_i, \quad i = 1,...100 \tag{24}$$

In this first case the data is noisy and uncorrelated with $\epsilon_i \sim N(0, 25)$ and is generated with $a = 2.5$ and $b = 1$. Using $N_{exp} = 80$ perfect modeling experiments (that is experiments where the model generates its own data), one can determine an informative prior for $S$ using equations in Section 4. In our case, these estimates gave $\alpha = 140$ and $\beta = 161$. Repeating this estimate with a different set of perfect modeling experiments will yield similar but not equivalent results. The priors for $a$ and $b$
are distributed uniformly, i.e. $p(a) \propto 1$ and $p(b) \propto 1$.

Assuming all data are independent, i.e. $k_e = 100$, Bayesian estimates of the optimal slope and intercept and their uncertainties are nearly equivalent to estimates based on least squares estimation (Figure 2; Table 1). The solutions are also nearly equivalent to estimates using non-informative values for $\alpha$ and $\beta$, i.e. $\alpha = 0$ and $\beta = 0$. The two estimates are the same since the log-likelihood and
the perfect modeling experiments assumed (correctly) that the data were uncorrelated data, resulting in similar posterior estimates of $S \sim 1$.

### 5.2 Correlated noise

We next consider the same problem but now where the data is affected by correlated noise. Estimates of the empirical Bayes parameters presumes that we have an adequate model of the uncertainties af-
fecting the data. For this example we use a reddening process with parameter $r$ to generate correlated noise $\eta$ from uncorrelated noise $\epsilon_i \sim N(0, 25)$,

$$\eta_i = r\eta_{i-1} + \sqrt{(1-r^2)}\epsilon_i, \tag{25}$$





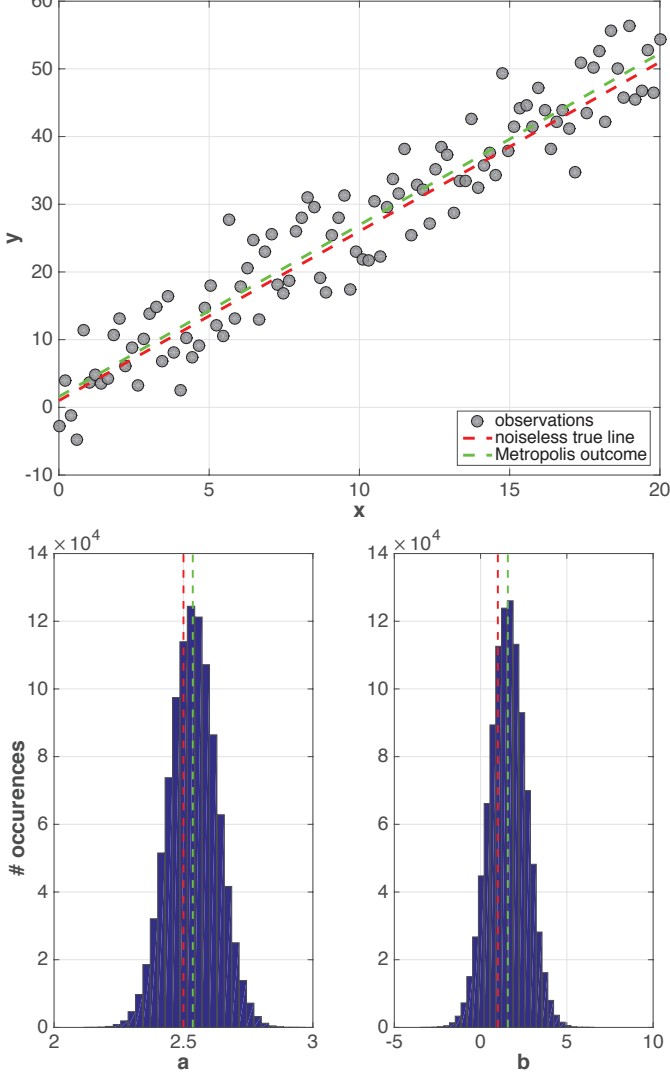

**Figure 2.** Example of fitting a line through noisy uncorrelated data (top) using $k_e = 100$, $\alpha = 140$, and $\beta = 161$. Also shown are solution marginal distributions for slope ($a$) and intercept ($b$). Although the estimated slope and intercept (green dashed lines) are slightly offset from their true values (red dashed lines), they are within the uncertainty of the fit. Solutions are nearly identical to those using mostly non-informative choices for prior on $S$, i.e. $p(S) \sim Ga(\alpha, \beta)$ with $\alpha = 0$, and $\beta = 0$.



**Table 1.** Results of different solution strategies for estimating slope ($a$) and intercept ($b$) when fitting a line through 100 data points with uncorrelated noise (equation 24). Estimates of optimal values for slope and intercept were nearly identical with $a = 2.54$ and $b = 1.58$. Similarly, uncertainties in these estimates were nearly identical whether the Bayes estimates used informed or uninformed priors or estimates were based on a least squares estimation.

| solution strategy | $k_e$ | $\alpha$ | $\beta$ | $\sigma_a$ | $\sigma_b$ | $<S>$ |
|---|---|---|---|---|---|---|
| Metropolis w/ unformative priors | 100 | 0 | 0 | 0.081 | 0.99 | 1.09 |
| Metropolis w/ informative priors | 100 | 140 | 161 | 0.084 | 0.98 | 1.10 |
| Least squares estimation | – | – | – | 0.081 | 0.94 | – |

where $i$ is an index of consecutive data points. The model with correlated noise is

$$d_i = ax_i + b + \eta_i. \tag{26}$$

Such a reddening process occurs naturally within climate data from the way the ocean integrates weather noise from the atmosphere, producing power at longer time scales. Figure (3) shows the same model as before but with noise drawn from a red noise process with $r = 0.8$ (equation 25). Correlations affect systematic changes in slope and intercept over a range of $x$ values (Figure 3). Thus we should expect that inferences of the slope and intercept are more uncertain.

Using $N_{exp} = 80$ perfect modeling experiments with a red noise process model affecting 100 data points, we estimate $k_e = 20$, $\alpha = 111$, and $\beta = 488$. Table 2 shows the uncertainties in the estimates for the slope and intercept using either informative or uninformative values for the effective degrees of freedom $k_e$ or values for $\alpha$, and $\beta$. Estimates using $k_e = 20$ using either informed or uninformed priors give comparable results, both indicating uncertainties are much larger when data are

correlated. The least squares estimation solution, while giving identical estimates of the slope and intercept, give much smaller estimates of the uncertainty comparable to estimates with the Metropolis algorithm that assumes data were uncorrelated (i.e. $k_e = 100$). The estimates of uncertainty from the least squares estimation are not large enough to include the model's actual slope and intercept, thus are too narrow.

The results show that the largest uncertainties come from the lower estimates of the effective degrees of freedom and uninformative priors for $S$. Including an informative prior for $S$ can reduce this uncertainty and still (barely) accommodate the true values of the slope and intercept (Figure 3). However this is not true for providing informative priors for $S$ without good information about the effective degrees of freedom. From this respect is it perhaps more important to provide good

information about the effective degrees of freedom than informative priors for $S$. The following section discusses one potentially useful strategy for making use of empirical estimates of $S$.





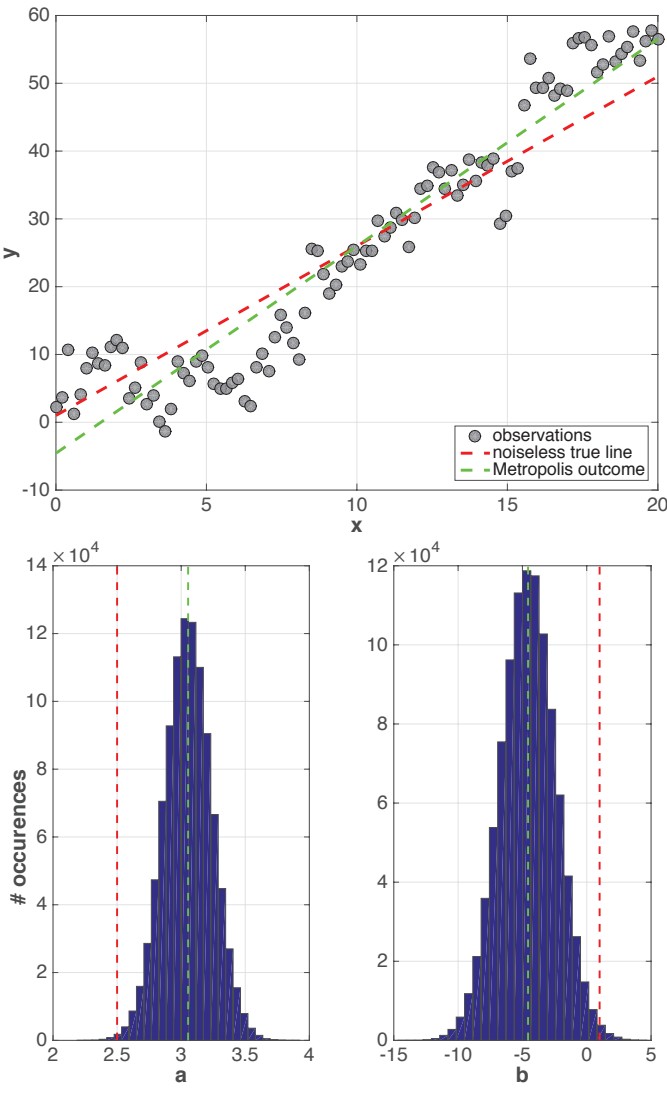

**Figure 3.** Example of fitting a line through data with correlated noise using Metropolis sampling and informative priors for $k_e = 20$, $\alpha = 111$, and $\beta = 488$. Also shown are solution marginal distributions for slope ($a$) and intercept ($b$). In contrast to the example with uncorrelated data (Figure 2) the estimated optimal parameter values (green dashed lines) are far from the true values (red dashed lines) and are only barely within estimated uncertainties.





**Table 2.** Estimates of uncertainties in estimating slope ($a$) and intercept ($b$) when fitting a line through 100 data points with correlated noise using a reddening process with $r = 0.8$. The estimates of the optimal values for $a$ and $b$ were the same for all solution strategies including least squares estimation with $a = 3.05$ and $b = -4.55$.

| solution strategy | $k_e$ | $\alpha$ | $\beta$ | $\sigma_a$ | $\sigma_b$ | $<S>$ |
|---|---|---|---|---|---|---|
| Metropolis w/ informative $k_e$ and $S$ | 20 | 111 | 488 | 0.19 | 2.15 | 0.22 |
| Metropolis w/ informative $k_e$ | 20 | 0 | 0 | 0.26 | 3.06 | 0.15 |
| Metropolis w/ informative $S$ | 100 | 111 | 488 | 0.16 | 1.87 | 0.27 |
| Metropolis w/ uninformative priors | 100 | 0 | 0 | 0.10 | 1.17 | 0.78 |
| Least squares estimation | – | – | – | 0.10 | 1.11 | – |

## 6  Component weighting

One of the main limitations of precision parameter $S$ as discussed above is that it can not affect the relative weighting among observational constraints. This relative weighting among observations is supposed to occur within a covariance matrix $\mathbf{C}$. Defining such a matrix becomes increasingly challenging as one expands the number of observables, regions, and seasons that typically are incorporated into climate model evaluation metrics. Here we describe a way one could choose to weight different components of a cost function within a multivariate normal framework using a separate $<S_q>$ for each quantity $q$.

Suppose one has created a set of $N_q$ model evaluation metrics $E(\mathbf{m})_q$ providing a normalized measure of distance between a model and observations for different fields, regions, or seasons as specified by index $q$. Summing these cost components assumes these measures are independent, which is not likely true. With $N_{exp}$ perfect modeling experiments one can generate separate estimates of $\{k_e\}_q$ and $<S_q>$ such that each component may be normalized separately and together such that the total cost function may be generated by

$$S_{tot}E(\mathbf{m})_{tot} = S_{tot}\sum_{q=1}^{N_q} <S_q> E(\mathbf{m})_q \tag{27}$$

where $p(S_{tot}) \sim Ga(\alpha, \beta)$. As formulated here, this process does not give any flexibility to adapt individual component weightings during sampling as the gamma priors only apply to $S_{tot}$. However this process does give opportunity to include correlation information, albeit that generated by the perfect model, into the total cost metric.

## 7  Application of $S$ to the calibration of CAM3.1

Although a precision parameter $S$ has been included in previous calibrations of CAM3.1 (Jackson et al., 2008), it was not fully explained nor did Jackson et al. (2008) evaluate the extent to which the outcome of those calculations were being affected by $S$. Here we present the results of a new





270    calibration that is related to the calculation reported in Jackson et al. (2008) in which Multiple Very
Fast Simulated Annealing (MVFSA) sampling (Jackson et al., 2004) is used to calibrate six param-
eters important to clouds, convection, and radiation within Community Atmosphere Model version
3.1 (CAM3.1) (Collins et al., 2006) with a resolution of 2.8° longitude by 2.8° latitude (T42) with
26 vertical levels. Important differences from the previous calculation include shorter 2-year rather

than 11-year model integrations with specified climatological sea surface temperatures and sea ice,
using only the last year for comparison to observations, and the use of ERA-40 (Uppala et al., 2005)
for vertically averaged (mass weighted) air temperature and humidity, zonal winds at 300 mb, and
sea level pressure, Willmott and Matsuura (2000) for 2-m air temperature over land, CERES2 (Loeb
et al., 2010) for long and shortwave cloud forcing, OAFlux data (Yu et al., 2008) for latent heat fluxes

over the ocean, and GPCP (Adler et al., 2009) for precipitation. Unlike Jackson et al. (2008), we did
not make use of any available data for low, medium, and high cloud amounts, estimates of short
and longwave radiation at the top of the atmosphere, or surface latent and sensible heat fluxes other
than what was mentioned above concerning OAFlux data. These changes were made to make the
calibration mimic the process that occurs within the NCAR Atmosphere Model Working Group for

evaluating the effects of different parameter choices as represented by the group's Diagnostics Pack-
age (https://www2.cesm.ucar.edu/working-groups/amwg/amwg-diagnostics-package) and "Top 10"
Taylor metrics. Similar to Jackson et al. (2008), we consider the same six parameters and same 30°
S to 30° N domain and seasonal (DJF, MAM, JJA, SON) averages for making model comparisons
to data as well as include a term in our cost function for global annual mean radiative balance. For

the updated calculation, we completed 2261 experiments over 16 independent MVFSA chains.

     The metrics for each of these observational targets consists of taking the spatial mean of grid point
differences divided by the spatial variance in the observational data. Global radiative balance was
the exception insofar as we were seeking solutions that were within a $1\,\mathrm{Wm^{-2}}$ of the target radiative
balance of $0.5\,\mathrm{Wm^{-2}}$. A set of 20 idealized experiments were used to estimate the variance for each

field and season which was used to weight each component of the cost according to equation 27.
The total cost calculation was given by the average of 41 cost components (ten quantities over four
seasons and the component for global radiative balance). The calculation does not include prior in-
formation about degrees of freedom, but does include prior information for the gamma parameters
using the 20 idealized experiments to get $\alpha = 3.45$ and $\beta = 0.0094$. The mean for the prior distri-

bution for $S$ is 367.0. While it would have been better to include prior information about degrees of
freedom, the dominant source of uncertainty likely comes from the inability of the model to match
all of the data simultaneously. $S$ captures this type of uncertainty and our purpose here is to highlight
this aspect of the calibration.

     The result of the calibration is an ensemble representing uncertainties in specifying seven param-

eters (six model parameters and $S$) such that the model is consistent with observational data. While
the number of ensemble members are relatively few (2261 samples), previous work has shown that





MVFSA sampling is particularly efficient at providing qualitative estimates of the posterior probability of the solution uncertainties (Jackson et al., 2008; Villagran et al., 2008). The marginal distributions for the six CAM3.1 model parameters and precision parameter $S$ as well as the distribution
of cost values for this ensemble is shown in Figure 4. The marginals are fairly broad, with many of the sixteen optimal parameter configurations covering a wide range of values. However that is not to say these parameters are unimportant. On the contrary, these parameters have a significant influence on simulated climates. The broad posterior distributions come about, in part, because of co-dependencies that arise during sampling model parameters to be consistent with observational
data (Table 3). For instance the critical relative humidity for high clouds ($rhminh$) which perhaps has the flattest distribution, is also the parameter that is the most strongly dependent on the choice of all the other parameters.

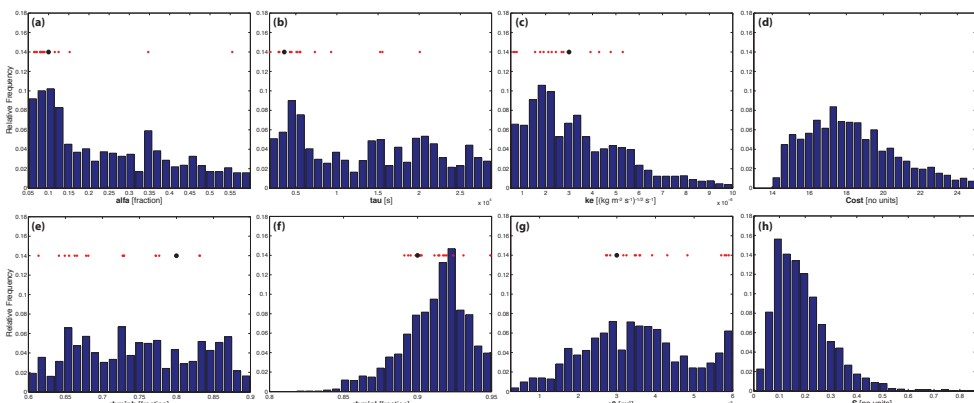

**Figure 4.** Marginals for 6 parameters within CAM3.1 important to clouds, convection, and radiation as well as for precision parameter S. Marginals are generated from 1948 samples of 2261 total which excludes experiments containing a cost greater than 25. Red dots indicate place in parameter space where each of the 16 chains were minimized. Black dot indicates the default model configuration. The parameters are *alfa* for the initial cloud downdraft mass flux, *tau* for the consumption rate of CAPE, *ke* for the environmental air entrainment rate, *rhminh* for the high cloud critical relative humidity, *rhminl* for the low cloud critical relative humidity, and *c0* for the precipitation efficiency.

**Table 3.** Sample correlation for CAM3.1 parameters within the posterior distribution. Parameters are defined within Figure 4.



|        | alfa | tau   | ke   | rhminh | rhminl | c0    |
|--------|------|-------|------|--------|--------|-------|
| alfa   | 1    | 0.35  | 0.17 | 0.31   | 0.15   | 0.09  |
| tau    | 0.35 | 1     | 0.02 | 0.22   | 0.12   | -0.07 |
| ke     | 0.17 | 0.02  | 1    | 0.12   | 0.17   | 0.09  |
| rhminh | 0.31 | 0.22  | 0.12 | 1      | -0.22  | 0.19  |
| rhminl | 0.15 | 0.12  | 0.17 | -0.22  | 1      | -0.09 |
| c0     | 0.09 | -0.07 | 0.09 | 0.19   | -0.09  | 1     |

The posterior mean value for $<S> = 0.19$ (Figure 4h) is quite a bit smaller from the prior mean value of $<S> = \frac{\alpha}{\beta} = 367$. Costs using observational data are quite a bit larger than those that occurred in the idealized experiments where model output was used as a surrogate for observational data. This has the effect of making the sampling algorithm more accepting of alternate model configurations and increasing the parametric uncertainties. Figure 5 illustrates which model configurations were accepted or rejected by different choices of parameters $rhminh$ and $rhminl$ which control the critical relative humidity for cloud formation for high clouds and low clouds, respectively. Estimates of the response surface, given by changes in cost values as a function of these two parameters, show that the parameter settings that allow CAM3.1 to have the lowest total cost (Figure 5a) are not the same for particular cost components such as column average relative humidity (Figure 5b), global radiative balance (Figure 5c), and shortwave cloud forcing (Figure 5d). The models that ended up being selected most frequently (and define the modes of the posterior distribution) represents a compromise. The sixteen model configurations representing separate optimizations (given by the red dots) are distributed near the center of these competing choices. All these configurations have a cost associated with their radiative balance of less than 50, which corresponds to models being within $4 \ \mathrm{Wm}^{-2}$ of the target radiative balance. So while the cost function for radiative balance specified acceptability to be models that were within $1 \ \mathrm{Wm}^{-2}$ of radiative balance, the high cost values and $S$ allowed the sampling algorithm to accept models with comparable skill that happen to include models with a larger energy imbalance.

## 8 Summary

The strength of observational evidence is a key part in testing hypotheses concerning the physics of climate change. Since much of the data that is used to evaluate hypotheses are correlated, it becomes important to include the effects of these correlations on model evaluation metrics to appropriately account for observational evidence when testing different model versions. Here we proposed a way to make use of perfect modeling experiments as a surrogate for missing observational data to estimate the effects of these correlations on climate model evaluation metrics. We introduce a precision parameter $S$ that uses empirical Bayes to specify informative values for information concerning the effective degrees of freedom $k_e$ and/or values for the scale and rate parameters $\alpha$ and $\beta$ for a





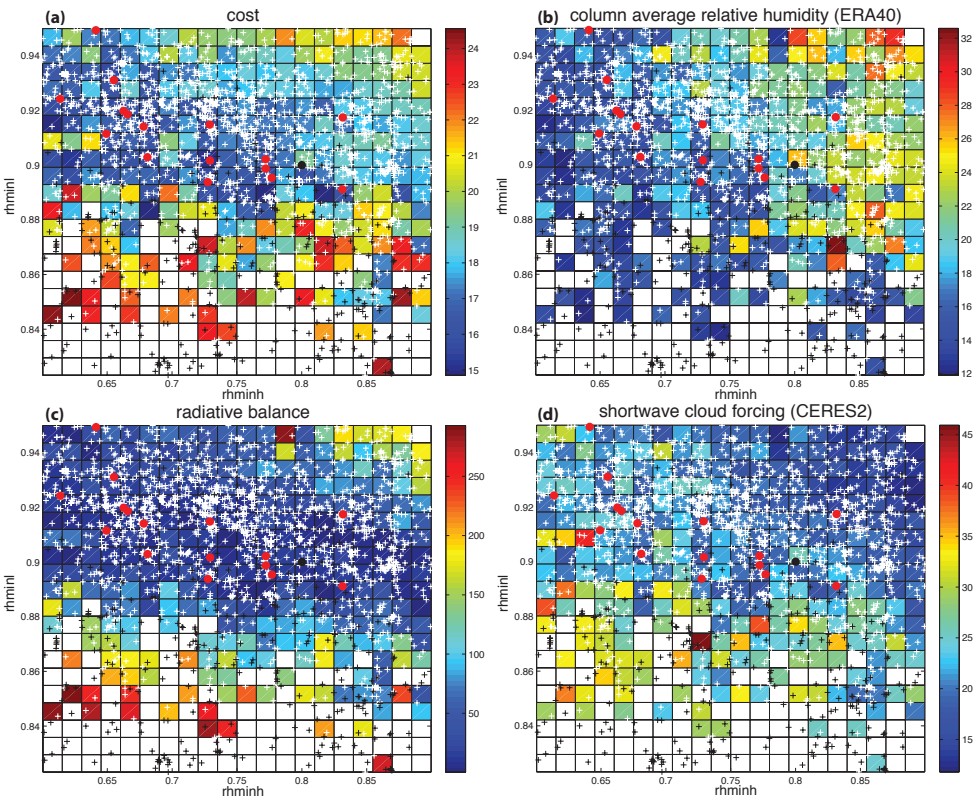

**Figure 5.** Estimates of the response surface for total cost (a), column average relative humidity (b), radiative balance (c), and shortwave cloud forcing (d). Each panel color shows a weighted averaged of experiments whose cost is less than 25 (given by the white '+') Jackson (2009). Experiments whose cost values are greater than 25 are indicated by a black '+' symbol. The sixteen red dots correspond to experiments whose cost were minimum within their respective sampling chains. The black dot corresponds to the default value of CAM3.1. The results of 2261 experiments are shown. All cost values have no units.





gamma prior distribution. We connected these concepts together to show how they work for a simple
example of fitting a line and discuss how these same concepts could be used to assign weights to
different sub-components of a model evaluation metric. This framework assumes errors are multi-
variate normal. The mean value for the precision parameter adapts to changes in size of the model
evaluation metric including model biases (or 'discrepancies' in Bayesian statistical terminology).
While assumptions of multivariate normal errors are often appropriate for long-term mean climate
data, no such assumption can be made for model biases. So while the use a precision parameter
is one way to expand uncertainties in proportion to model biases, prediction errors are not related
in any obvious way to commonly proposed climate model evaluation metrics (Klocke et al., 2011;
Masson and Knutti, 2013; Sanderson and Sanderson, 2013; Caldwell et al., 2014). While some have
proposed including a statistical model that compensates biases in model predictions (Brynjarsdottir
and O'Hagan, 2014), it is not clear how this can be accomplished for high-dimensional systems. One
of the advantages of using precision parameter $S$ is that its interpretation is fairly straightforward
mathematically and scientifically. This makes it a good starting point to think about how to address
even more challenging aspects of quantifying uncertainties in climate projections using observational
data.

## 9 Code availability

Matlab and R code for generating the results shown in figures (2) and (3) may be found at Jackson
and Huerta (2015).

*Acknowledgements.* This material is based upon work supported by the U. S. Department of Energy Office of
Science, Biological and Environmental Research Regional & Global Climate Modeling Program under Award
Numbers DE-SC0006985 and DE-SC0010843. The authors would like to thank Michael Tobis for developing
and running the python scripts that managed the sampling of CAM3.1 parametric uncertainties on the Lones-
tar4 HPC resource at the UT Texas Advanced Computing Center. The portion of this work related to estimating
CAM3.1 parametric uncertainties used allocation award ATM100049 from the Extreme Science and Engineer-
ing Discovery Environment (XSEDE) program, which is supported by the National Science Foundation.





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
