# Peer review of "Empirical Bayes approach to climate model calibration"

_Geoscientific Model Development, 2016_

## Referee Comment (RC1) · Anonymous Referee #1 · 20 May 2016

Review of "Empirical Bayes approach to climate model calibration" by CS Jackson and G Huerta

**General recommendation**

The topic of the paper is model calibration and the new thing it seems to introduce is the scaling factor $S$. While this is of general interest for GMD readers, the paper should not be published in its present form. I have several reasons for this:

1) It is not clear to me what is new in this manuscript relative to the older one by the same authors (Jackson et al, 2008) where the scale factor is already introduced.

[Figure]

2) The reason why this remains unclear is that the paper is so poorly written that almost everything remains unclear. In fact, I stopped reading after section 4 because at that point I still did not have any clue about what the authors intend to develop.

3) The mathematics used lacks clear definitions (see below) which is probably the main reason why much of the paper is non-understandable. I found that I was always guessing what the authors intended to say. Furthermore it seems that there are inconsistencies (see below).

4) References to similar or related work, in particular to introductory texts on model calibration, are missing. There is no mentioning, let alone discussion, of previous work (except from the same group), no comparison with other approaches, no pros or cons.

For all these reasons I suggest to reject this paper.

**Major comments**

1) Although background material on model calibration is provided in Sect. 2, it does not suffice to my feeling. Readers without experience in model calibration want to have a more basic introduction or alternatively references to more basic introduction.

Certainly, there is some literature on this topic and it is not good that (almost) nothing is cited in the introduction. It should be stated what is novel, different, better, etc., in the presented method in comparison with other approaches.

2) The mathematical introduction in section 2 is unclear in various respects and insufficient.

a) It is unclear whether $d$ means either a quantity like "temperature", or a specified value of that quantity, as "300K". In the latter case, is it the "true" value of the observable or a measured datum subject to measurement error? Accordingly it is unclear whether $\hat{x}$

is an estimate of the true value or simply a model result for an observable $d$, that differs from the measured $d$. In the first case, the residuum $\epsilon$ can be viewed as a random quantity, but in the second case it is given as $d - \hat{x}$ which in turn are both given as well, thus $\epsilon$ should be non-random in that case. I guess that the probability space is the space of parameter vectors $m$, but that is not stated as well.

b) If I accept that $\epsilon$ is a gaussian random quantity, I still do not see why its expectation value should be zero, in particular when there is missing physics in the model which could easily produce a bias.

c) As the meaning of both $d$ and $x$ remain unclear, it is unclear what eq. 1 actually means. Is it, as a function of parameter vector $m$, the probability that $\hat{x}$ comes out as $d$?

d) To my view the chain of arguments gets broken where the authors mention that the covariance matrix can be rank deficient. Here they enter into a side topic (eofs) that does not lead to the goal, which I think is to demonstrate how covariance leads to problems in model calibration. And to my opinion, the latter goal is not reached at all.

e) Finally, if $\epsilon \sim N(0, \sigma^2)$, then of course $\epsilon^2/\sigma^2 \sim \chi^2$ (equation 6). In this sense, the argument under the exponential function of a gaussian distribution is always $\chi^2$ distributed. In a similar way one could say that the log of $\epsilon$ is log-normally distributed. It is true, but I do not understand why this is stated here. Further I do not understand why the mean and variance of the $\chi^2$ are important instead of the mean and variance of the gaussian.

3) Section 3.

a) What is a "climate model evaluation matrix"? As it may be arbitrarily defined, how is it related to the covariance matrix in section 2?

b) 2nd sentence: I don't see the argument. I understand that information on model matching data can be used to determine parameter uncertainties, but how does this

statement follow from the first part of the sentence, namely, that statisticians often use a scaling factor in calibration?

c) Please explain how the likelihood function of eq. 11 can be a gaussian although $S$ is not considered a constant. If it is not a gaussian, then it is questionable whether a gamma distribution for $S$ is still a conjugate prior.

d) How can it be explained that the covariance matrix is suddenly reduced to a diagonal matrix in equation 13. Is this because of the EOF transformation? How can the reader see this in equations 12 and 13?

4) Section 4.

In Section 2 $k_e$ was introduced as the effective degrees of freedom, following from an EOF decomposition of the modelled fields. This gives the reader the impression that the determination of $k_e$ is relatively straightforward. Now, in section 4.1., nothing remains clear. It is not clear where the EOF decomposition is in this derivation. The factor $A/2$ is introduced seemingly without necessity, because it is already gone in eq. 19, just after its introduction in eq. 18. It is not at all clear why and how the number of experiments affects $k_e$.

**Minor problems**

1) Why is there a section 2.1 when there is no section 2.2?

2) A square root is missing in eq. 2.

3) Misuse of the "=" sign in eq. 6 and in eq. 18

4) line 99: replace phrase "probabilities are narrower".

5) Check brackets in eq. 11.

---

## Editor Comment (EC1) · K. Gierens (Editor) · 14 Jul 2016

I got a second opinion per email. The essential points are quoted here.

This paper, which is statistical in nature, does not achieve the level of clarity which would be appropriate .... Moreover, the authors' statistical model used to link simulator output and observations is ... inverted compared to the dominant model in the statistical field of Computer Experiments, which asserts the existence of a 'best' parameterisation $m^*$ for which $d|m^* = g(m^*) + \epsilon + e$ where $\epsilon$ is a contribution from structural uncertainty and $e$ is a contribution from measurement error, both multivariate (the dependence of $\epsilon$ on $m^*$ is usually suppressed). Because the characteristics of $\epsilon$ and $e$ are rather different, it pays to separate them; $e$ in particular

may have a zero expectation and diagonal variance matrix. This model has been dominant for nearly 20 years, and it is the standard model within which we might consider 'calibration', which is finding the conditional distribution $m^*|d$, on the basis of a prior distribution for $m^*$.

By contrast, the authors' statistical model is not even clear, because line 54 and eq (1) are incompatible, and the idea of the model correlating the observations makes little sense – observations do not care about models! I think the authors go on to adopt the standard model given above, with (5) being the likelihood function under a fully-Gaussian model for $\epsilon + e$. But there is no $m$ on the righthand side of (5), and $x$ is never properly defined, so it is hard to say. I think the authors are saying that we should include an uncertain scale parameter in the covariance function of $\epsilon$. This is hardly innovative....

There have been big problems to understand what the authors are doing. I must admit, I share these problems. And I think it is not just because I am not a specialist in model calibration. To my view, a statistical topic like model calibration should be interesting to readers of GMD, i.e. model developers. But typically they are not experts in advanced statistical topics. To my view, it is not necessary that a general reader understands all details, but the text must be clear so that the general idea (what is it good for and how could I try to use it) becomes comprehensible.

The outcome of the review process is that both reviewer's recommend rejection, in particular because of lack of clarity. I will follow these recommendations and reject this paper.

---

## Author Comment (AC1) · 15 Aug 2016

*Response to reviewer # 1 and #2 input on*
**"Empirical Bayes approach to climate model calibration" by Jackson and Huerta**

August 11, 2016

**Response to Reviewer # 1** Reviewer comment given in blue.

General recommendation The topic of the paper is model calibration and the new thing it seems to introduce is the scaling factor S. While this is of general interest for GMD readers, the paper should not be published in its present form. I have several reasons for this: 1) It is not clear to me what is new in this manuscript relative to the older one by the same authors (Jackson et al, 2008) where the scale factor is already introduced. 2) The reason why this remains unclear is that the paper is so poorly written that almost everything remains unclear. In fact, I stopped reading after section 4 because at that point I still did not have any clue about what the authors intend to develop. 3) The mathematics used lacks clear definitions (see below) which is probably the main reason why much of the paper is non-understandable. I found that I was always guessing what the authors intended to say. Furthermore it seems that there are inconsistencies (see below). 4) References to similar or related work, in particular to introductory texts on model calibration, are missing. There is no mentioning, let alone discussion, of previous work (except from the same group), no comparison with other approaches, no pros or cons. For all these reasons I suggest to reject this paper.

We thank the reviewer for his or her time to review the manuscript.

The objective of the paper concerns an aspect of Bayesian calibration related to inclusion of a precision parameter S. We did not try to introduce or explain Bayesian calibration which, in retrospect, may have been helpful to provide.

We first want to point out that while the results of Jackson et al., (2008) include an uncertain precision parameter 'S' to account for uncertainties in the covariance matrix, Jackson et al. (2008) "defers to a future work to discuss the details of the treatment of S in our approach to quantifying the effects of observational and other sources of uncertainty in our estimates of parametric uncertainties." In that

paper, Jackson and co-authors do not present the results on their estimates of 'S' nor discuss how that parameter affected the calibration outcomes. Inclusion of such a parameter is common practice within Bayesian calibration, thus its inclusion should not be considered novel then or now. What prompted the writing of the current manuscript is our deepening understanding of its interpretation, both its strength and limitations, and a more thorough look at how it affects calibration results. More importantly, we had previously failed to recognize the importance of a parameter for the effective degrees of freedom, which we we show in the present manuscript is critical to correctly estimating the statistical significance of model-data differences. What we provide that is new in this regard is a way to take an ensemble of model output to estimate the effective degrees of freedom as well as provide a rationale for setting values to the parameters for the gamma distribution priors. Since we are using 'data' to inform the prior, this empirical approach is considered non-standard within Bayesian inference which typically eschews such approaches.

We were remiss in not mentioning the small but growing work in climate model calibration. There are a few papers that use Bayesian inference for climate model calibration (e.g. Rougier 2007; Jarvinen et al., 2010; Hakkarainen et al., 2012; Hauser et al., 2012). These papers do not explicitly use or call attention to a precision parameter. There are other papers that deal with non-Bayesian approaches to climate model calibration (e.g. Neelin et al, 2010; Rowlands et al, 2012; Mauritsen et al, 2012; Shiogama et al., 2013; Schirber et al. 2013).

1. Although background material on model calibration is provided in Sect. 2, it does not suffice to my feeling. Readers without experience in model calibration want to have a more basic introduction or alternatively references to more basic introduction. Certainly, there is some literature on this topic and it is not good that (almost) nothing is cited in the introduction. It should be stated what is novel, different, better, etc., in the presented method in comparison with other approaches.

We agree some introductory material on Bayesian calibration would be helpful.

2. The mathematical introduction in section 2 is unclear in various respects and insufficient. a) It is unclear whether $d$ means either a quantity like "temperature", or a specified value of that quantity, as "300K". In the latter case, is it the "true" value of the observable or a measured datum subject to measurement error? Accordingly it is unclear whether $\hat{x}$ is an estimate of the true value or simply a model result for an observable $d$, that differs from the measured $d$. In the first case, the residuum $\varepsilon$ can be viewed as a random quantity, but in the second case it is given as $d - \hat{x}$ which in turn are both given as well, thus $\varepsilon$ should be non-random in that case. I guess that the probability space is the space of parameter vectors $m$, but that is not

stated as well.

In the example we were dealing mainly with modeling error. So $d$ may be considered a true value with model estimate $\hat{x}$ and error $\varepsilon$.

b) If I accept that $\varepsilon$ is a gaussian random quantity, I still do not see why its expectation value should be zero, in particular when there is missing physics in the model which could easily produce a bias.

Quite true. In this simple example, we are providing background concerning the use of a covariance matrix to account for dependencies between two observables. This becomes useful in our explanation of the limits of using of a single precision parameter to 'correct' for deficiencies in the specification of a covariance matrix. The point you bring up is important and we discuss it within the final section of the paper as any biases will be included in estimates of $S$ through the log-likelihood $E$.

c) As the meaning of both $d$ and $x$ remain unclear, it is unclear what eq. 1 actually means. Is it, as a function of parameter vector $\mathbf{m}$, the probability that $\hat{x}$ comes out as $d$?

yes

d) To my view the chain of arguments gets broken where the authors mention that the covariance matrix can be rank deficient. Here they enter into a side topic (eofs) that does not lead to the goal, which I think is to demonstrate how covariance leads to problems in model calibration. And to my opinion, the latter goal is not reached at all.

On the contrary, we think it is useful to think about the log-likelihood and the multivariate normal distribution as a test statistic with $k_e$ degrees of freedom. Here we connect $k_e$ to the well established notion in climate data analysis where it is common to limit comparisons between models and data to the $eofs$ with the largest eigenvalues. We want to connect the use of the precision parameter $S$ to a lack of knowledge of how many $eofs$ are unique given the dependencies that exist among the data used to test the model. The common assumption in climate model calibration is that all the data are independent.

e) Finally, if $\varepsilon \sim N(0, \sigma^2)$, then of course $\frac{\varepsilon^2}{\sigma^2} \sim \chi^2$ (equation 6). In this sense, the argument under the exponential function of a gaussian distribution is always $\chi^2$ distributed. In a similar way one could say that the log of $\varepsilon$ is log-normally distributed. It is true, but I do not understand why this is stated here. Further I do not understand why the mean and variance of the $\chi^2$ are important instead of the mean and variance of the gaussian.

In section 4.1, we use the mean and variance of a distribution of cost values to estimate the effective degrees of freedom as well as values for the two parameters

needed for the gamma distribution prior. In particular, in a perfect model situation where the target observations make use of model output, the distribution of cost values generated from internal variability will be distributed like a $\chi^2$ distribution with $k_e$ degrees of freedom.

3. Section 3. a) What is a "climate model evaluation matrix"? As it may be arbitrarily defined, how is it related to the covariance matrix in section 2?

Sorry, this was poorly worded. We think of precision parameter S as a factor multiplying the covariance matrix within the log-likelihood. It is the same covariance matrix we were working with in section 2.

b) 2nd sentence: I dont see the argument. I understand that information on model matching data can be used to determine parameter uncertainties, but how does this statement follow from the first part of the sentence, namely, that statisticians often use a scaling factor in calibration?

What we wrote was not clear. Here we introduce the use of a precision parameter S to account for uncertainties in the specification of C. This is commonly done within Bayesian calibration. In particular one can update S using information about the scatter between a model and data.

c) Please explain how the likelihood function of eq. 11 can be a gaussian although S is not considered a constant. If it is not a gaussian, then it is questionable whether a gamma distribution for S is still a conjugate prior.

The likelihood function conditional on m looks like a Gamma distribution for S, this is why the gamma prior on S works as conjugate in the Gibbs Sampling scheme.

d) How can it be explained that the covariance matrix is suddenly reduced to a diagonal matrix in equation 13. Is this because of the EOF transformation? How can the reader see this in equations 12 and 13?

There is a mistake with this equation. What it should be is

$$
|S^{-1}\mathbf{C}|^{\frac{1}{2}} = \det\left( \begin{bmatrix} \frac{1}{S} & & 0 \\ & \ddots & \\ 0 & & \frac{1}{S} \end{bmatrix}_{[k_e \times k_e]} \right)^{\frac{1}{2}} |\mathbf{C}|^{\frac{1}{2}} = (\tfrac{1}{S})^{\frac{k_e}{2}} |\mathbf{C}|^{\frac{1}{2}}. \tag{1}
$$

4) Section 4. In Section 2 ke was introduced as the effective degrees of freedom, following from an EOF decomposition of the modelled fields. This gives the reader the impression that the determination of ke is relatively straightforward. Now, in section 4.1., nothing remains clear. It is not clear where the EOF decomposition

is in this derivation. The factor A/2 is introduced seemingly without necessity, because it is already gone in eq. 19, just after its introduction in eq. 18. It is not at all clear why and how the number of experiments affects ke.

The EOF decomposition does not necessarily tell you what $k_e$ should be. More importantly many cost functions being used by climate scientists do not lend themselves to EOF decomposition as they often involve arbitrary number of fields, regions, and seasons. The early sections of the manuscript were making an illustration of how dependencies affect the degrees of freedom. Here we show an alternate way to estimate the effective degrees of freedom using a ensemble of cost values that may be arbitrarily defined.

The factor of A is there to emphasize that the distribution of cost values as we have defined it will only be proportional to a $\chi^2$ distribution. A is the constant of proportionality. We show that A drops out.

Minor problems

1. Why is there a section 2.1 when there is no section 2.2?

   Thanks for pointing this out.

2. A square root is missing in eq. 2.

   Thanks for finding that error.

3. Misuse of the "=" sign in eq. 6 and in eq. 18

   Ok

4. line 99: replace phrase "probabilities are narrower".

   Ok

5. Check brackets in eq. 11.

   They look ok.

**Response to Reviewer # 2** Reviewer comment given in blue.
This paper, which is statistical in nature, does not achieve the level of clarity which would be appropriate .... Moreover, the authors' statistical model used to link simulator output and observations is ... inverted compared to the dominant model in the statistical field of Computer Experiments, which asserts the existence of a 'best' parameterisation m* for which d — m* = g(m*) + epsilon + e where epsilon is a contribution from structural uncertainty and e is a contribution from measurement error, both multivariate (the dependence of epsilon on m* is usually suppressed). Because the characteristics of epsilon and e are rather different, it pays to separate

them; e in particular may have a zero expectation and diagonal variance matrix. This model has been dominant for nearly 20 years, and it is the standard model within which we might consider 'calibration', which is finding the conditional distribution m* — d, on the basis of a prior distribution for m*.

By contrast, the authors' statistical model is not even clear, because line 54 and eq (1) are incompatible, and the idea of the model correlating the observations makes little sense – observations do not care about models! I think the authors go on to adopt the standard model given above, with (5) being the likelihood function under a fully-Gaussian model for epsilon + e. But there is no m on the righthand side of (5), and x is never properly defined, so it is hard to say. I think the authors are saying that we should include an uncertain scale parameter in the covariance function of epsilon. This is hardly innovative....

We thank the reviewer for sharing his or her perspectives. We are familiar with the statistical literature and discussion surrounding the treatment systematic errors, epsilon, between the model and data. We intentionally do not include it here as we are unaware of any decent way to include it within climate model calibration. We discuss the need for its treatment in section 8 and how its neglect affects the interpretation of precision parameter S.

We also acknowledge that inclusion of a precision parameter S is not new. What is new is our approach to making use of climate model output to help determine the effective degrees of freedom $k_e$ as well as the two parameters that define the gamma distribution prior for S. Since nearly all of these concepts are new to climate scientists, one of the goals of our manuscript was explain how all this works using a simple example of fitting a line through a set of points.

We stand by our description of the simple model in section 2 that makes use of equation (1) that follows from those in line 54 insofar as typical climate models $g(\mathbf{m}) = \hat{x}$ can only provide statistical estimates of observables $d$. That is because climate models are turbulent and one can not expect model output to exactly match observations. Moreover, climate scientists often use climate models to estimate the kinds of correlations that exist in data. This is the central idea of our 'empirical' Bayes approach to this problem. We are using the output of a set of experiments to learn about the effective degrees of freedom that exist within our data so that we can inform the prior distributions for the precision parameter S.